

# Endophytic actinomycetes promote growth and fruits quality of tomato (*Solanum lycopersicum*): an approach for sustainable tomato production

Jeeranan Khomampai[1], Nakarin Jeeatid[1], Thewin Kaeomuangmoon[1], Wasu Pathom-aree[2], Pharada Rangseekaew[2], Thanchanok Yosen[3], Nuttapon Khongdee[4] and Yupa Chromkaew[1]

[1] Department of Plant and Soil Sciences, Faculty of Agriculture, Chiang Mai University, Chiang Mai, Thailand
[2] Department of Biology, Faculty of Science, Chiang Mai University, Chiang Mai, Thailand
[3] Central Laboratory, Faculty of Agriculture, Chiang Mai University, Chiang Mai, Thailand
[4] Department of Highland Agriculture and Natural Resources, Chiang Mai University, Chiang Mai, Thailand

Corresponding authors
Nuttapon Khongdee,
nuttapon.k@cmu.ac.th
Yupa Chromkaew, yupa.c@cmu.ac.th

## ABSTRACT

**Background**. Tomato, a fruit with a high vitamin content, is popular for consumption and economically important in Thailand. However, in the past year, the extensive usage of chemicals has significantly decreased tomato yields. Plant Growth-Promoting Rhizobacteria (PGPR) is an alternative that can help improve tomato production system growth and yield quality while using fewer chemicals. The present study aimed to determine whether endophytic actinomycetes promote growth and fruit quality of tomato (*Solanum lycopersicum*).

**Methods**. The experiment was conducted in a net-houses at the Center for Agricultural Resource System Research, Faculty of Agriculture, Chiang Mai University, Chiang Mai province, Thailand. The randomized completely block design (RCBD) was carried out for four treatments with three replications, which was control, inoculation with TGsR-03-04, TGsL-02-05 and TGsR-03-04 with TGsL-02-05 in tomato plant. Isolated *Actinomycetes* spp. of each treatment was then inoculated into the root zone of tomato seedlings and analyzed by Scanning Electron Microscopy (SEM). The height of tomato plants was measured at 14, 28, 56, and 112 days after transplanting. Final yield and yield quality of tomato was assessed at the maturity phase.

**Results**. The SEM result illustrated that the roots of tomato seedlings from all treatments were colonized by endophytic actinomycetes species. This contributed to a significant increase in plant height at 14 days after transplanting (DAT), as found in the TGsR-03-04 treatment (19.40 cm) compared to the control. Besides, all inoculated treatments enhanced tomato yield and yield quality. The highest weight per fruit (47.38 g), fruit length (52.37 mm), vitamin C content (23.30 mg 100 g$^{-1}$), and lycopene content (145.92 µg g$^{-1}$) were obtained by inoculation with TGsR-03-04. Moreover, the highest yield (1.47 kg plant$^{-1}$) was obtained by inoculation with TGsL-02-05. There was no statistically significant difference in the number of fruits per plant, fruit width, brix, and antioxidant activity when various inoculations of endophytic actinomycetes were applied. Therefore, the use of endophytic actinomycetes in tomato cultivation may be an alternative to increase tomato yield and yield quality.

# INTRODUCTION

Tomatoes, scientifically known as *Lycopersicon esculentum Mill.*, belong to the Solanaceae family and are renowned for their widespread consumption and significant commercial value. Vegetables provide versatility as they can be consumed in their raw state, processed, or utilized in many industries. Moreover, tomatoes provide a high concentration of essential minerals, including vitamins, lycopene, and antioxidants, which confer advantageous effects on human health. In Thailand, tomato production is experiencing annual growth, however, the excessive use of chemicals leads to a decline in the efficacy of plant nutrients and a reduction in the availability of nutrients in the soil. Therefore, there is a need for improving tomato quality as well as reducing chemical utilization. In the recent literature, microorganisms are an approach to enhance plant growth that has gained significant attention. It has been demonstrated that beneficial microbes are involved in the fixation of atmospheric nitrogen, the breakdown of organic wastes and residues, the detoxification of pesticides, the inhibition of soil-borne pathogens and plant diseases, the improvement of nutrient cycling, and the synthesis of bioactive substances like vitamins, hormones, and enzymes that stimulate plant growth (*Adeniji, Aremu & Babalola, 2019*; *Omotayo & Babalola, 2021*). One of the example soil microorganisms is actinomycetes. The use of actinomycetes promoted the growth of tomato plants in terms of height, fresh weight of shoot and root including root length (*El-Tarabily et al., 2009*) as well as increased the content of lycopene and antioxidants in tomatoes (*Inculet et al., 2019*). Actinomycetes are gram-positive bacteria belonging to the order Actinomycetales, classified within the class Actinobacteria and subclass Actinobacteridae. They are distinct from other gram-positive bacteria due to their high guanine and cytosine content (57–75 mol%). The radial mycelium of classic actinomycetes is well developed and can be further classified into substrate and aerial mycelium based on their form and function. Complex structures including spores, spore chains, sporangia, and sporangiospores can be formed by certain actinobacteria and can be seen by microscopy (*Li et al., 2016*). Symbiosis refers to the ecological relationship where an endophytic actinomycete cohabits with its host plant, resulting in mutual dependence and benefits for both parties. In this relationship, endophytes receive nourishment and habitat from the host plants that harbor them. These endophytic actinomycetes, in turn, produce significant metabolites that protect the host plants from environmental stress, enhance their resistance to diseases and pests, and improve their adaptability to various environmental conditions (*Bacon & White, 2000*). Colonization into the root of endophytic species has a specific pattern and area for each endophytic species (*Zachow et al., 2010*). The endophyte begins to penetrate into the tissues of the host plant in the spaces between the plant cells, the endophytes exist in gaps between the mesophyll and parenchyma layers of the host plant where the organism lives are filled with mycelium. They are spread over the entire host plant without making the plant exhibit symptoms of disease (*Panphut, 2001*). *Adedayo et al. (2022)* revealed that actinomycetes

have a growth-promoting mechanism by producing hormones used to stimulate the growth of beneficial microorganisms around plant roots as well as the capacity to dissolve phosphorus in soil. It allows fixed phosphorus in soil to be dissolved and used by plants. Moreover, actinomycetes possess the remarkable ability to thrive in hostile environments and produce phytohormones, which support and stimulate plant growth. In addition to these growth-promoting substances, they also produce antibiotics, further highlighting their significant role in enhancing plant health and resilience. Numerous studies have shown that endophytic actinomycetes, particularly *Bacillus* spp., possess the potential to support plant growth. However, research focusing on the effects of *Streptomyces* and *Nocardiopsis* in enhancing growth and fruit quality in tomatoes is comparatively limited. Given this context, we propose the hypothesis that endophytic actinomycetes can reduce the excessive use of chemical fertilizers, increase yields, and have no negative effect on nutrients in plants and soil. Furthermore, they are capable of promoting both the growth and fruit quality of tomatoes. The primary objective of this study is to investigate how each endophytic actinomycetes influence the growth and fruit quality of tomato plants (*Solanum lycopersicum*).

## MATERIALS & METHODS

### Experimental designs

The experiment was conducted from October 2021 to March 2022 in a net-houses at the Center for Agricultural Resource System Research, Faculty of Agriculture, Chiang Mai University, Muang District, Chiang Mai Province, Thailand. The study was designed using a Randomized Complete Block Design (RCBD) to evaluate four treatments across three replications. These treatments included a control (non-inoculated with actinomycetes), inoculation with TGsR-03-04, inoculation with TGsL-02-05, and a combined inoculation of TGsR-03-04 with TGsL-02-05.

### Preparation of actinomycete endophytes

Isolated endophytic actinomycetes were prepared according to *Shutsrirung et al. (2013)* by selecting two isolates of actinomycetes, *Streptomyces violaceorectus* (TGsR-03-04) and *Nocardiopsis alba* (TGsL-02-05), which have the capacity to produce Indole-3-acetic acid (IAA) and dissolve phosphorus and potassium as well as act as plant growth promoting rhizobacteria (PGPR). After that, inhibitory mold agar 2 (IMA-2) medium was used to grow the endophytic actinomycetes, which were subsequently incubated at 30 °C for 7 days as well as shaking it for 7 days at 120 rpm at room temperature.

### Tomato seed germination and cultivation

Tomato seeds (F1 gamma of East West Seed, Thailand) were used in this study. The seed coating was removed, and the seeds were placed in distilled water for three minutes. Then, the seeds were wrapped in a thin white towel to sterilize their surface. The seeds were soaked in a 3% sodium hypochlorite solution for 1 min and rinsed three times with sterile distilled water for 3 min each time. After that, the seeds were placed on filter paper until dry. The seeds were sown on a plate covered with sterile filter paper and distilled water and

**Table 1  Chemical properties of growing material used in this study.**

| | pH | EC (dS/m) | OM (%) | Plant nutrients (%) | | | | |
|---|---|---|---|---|---|---|---|---|
| | | | | N | P | K | Ca | Mg |
| Growing material | 7.4 | 0.64 | 15.59 | 0.78 | 0.08 | 0.76 | 1.35 | 0.04 |

incubated in the dark for 2–3 days or until the tomato seeds had 0.5 cm long roots. Then, the tomato seedlings were sown in growing media (peat moss: rice husk: coconut coir compost 1:1:0.5, v/v) in a seedling tray. Actinomycetes with a spore concentration of 107 cfu/ml, 1 ml per plant, were inoculated at 15 days and after 30 days after planting the seedlings. The seedlings were transplanted separately into pots (8 × 13 inch) containing 5 kg of the growing media (coconut coir compost: soil: rice husk 1:1:1, v/v) in a net house. The spacing was 80 cm ×60 cm. The initial soil properties used for this experiment were analyzed as shown in Table 1. The analyzing protocols followed standard soil analysis, including pH by pH meter (*Corwin & Rhoades, 1984*), electrical conductivity by conductivity meter (*National Soil Survey Center, 1996*), organic matter (OM) titrated with ferrous sulfate (*Nelson & Sommers, 1980*), total N by colorimetry method (*Novozamsky et al., 1974*), total P by spectrophotometer (*Suwanwong, 2001*), and total K, Ca, and Mg by atomic absorption spectrophotometer (*Kalra, 1998*).

## Plant management

After transplanting, tomato plants received 0.6 liters of water per plant through drip irrigation. Seven days post-transplantation, a compound fertilizer with an NPK ratio of 16-16-16 was applied at a rate of 10 grams per plant. A second application of the same compound fertilizer was made 15–20 days after the first, also at 10 grams per plant. Additionally, a water-soluble fertilizer with a ratio of 0-0-60 was mixed with calcium nitrate (15-0-0), using 24 g of the fertilizer and 32 g of calcium nitrate dissolved in 44 liters of water. This solution was then applied to the plants twice a week at a rate of 0.3 liters per plant. From the reproductive stage to harvest, the fertilization was maintained with the compound fertilizer 16-16-16 applied every 15–20 days at 10 g per plant. The water-soluble fertilizer (0-0-60) and calcium nitrate mixture was continued at the same rate and frequency as before. Furthermore, a solution containing calcium boron (CaO: 31%, Zn: 1.4%, B: 0.1% w/w) was mixed with a magnesium solution (MgO: 31% w/w) at a rate of 10 ml each per 20 liters of water. This mixture was sprayed on the plants once a week at a rate of 20 ml per plant, providing essential nutrients for optimal growth and development.

## Root colonization analysis

Root seedlings inoculated with TGsR-03-04, TGsL-02-05, and a combination of TGsR-03-04 + TGsL-02-05 were examined under a Scanning Electron Microscope (SEM: JSM IT300). Small root segments from each treatment group were initially fixed with 2.5% glutaraldehyde in 0.1 M phosphate buffer, followed by rinsing with 0.2 M phosphate buffer at pH 7.2. The specimens were then post-fixed in 1% osmium tetroxide ($OsO_4$) in the same buffer. To prepare for SEM imaging, the fixed specimens underwent a gradual dehydration

process through a series of ethanol solutions of increasing concentrations (30%, 50%, 70%, 80%, 90%, and finally 100%). After dehydration, the specimens were subjected to critical point drying using liquid carbon dioxide ($CO_2$) in a pressure chamber, a step that preserves their structural integrity for detailed observation under the SEM.

## Plant sampling

Tomato plant height was monitored bi-weekly for four months following transplantation. During each harvest period, tomato fruit samples were systematically collected to evaluate the productivity and quality of the yield. Specifically, samples were taken from five plants per treatment, across three replications, with the data being recorded in terms of the number of fruits per plant and the yield per plant. For a detailed analysis of fruit quality, tomatoes were randomly selected from each treatment group, with a total of 30 fruits per treatment being examined. The quality data recorded for these fruits included weight per fruit, fruit width, fruit length, and Brix degree (a measure of sugar content).

Additionally, at the second harvest, 15 fruits per treatment were randomly chosen for a more focused analysis of their nutritional content, including measurements of vitamin C, antioxidants, and lycopene levels. This comprehensive approach provided a well-rounded understanding of the effects of the different treatments on both the quantity and quality of the tomato yield.

## Vitamin C analysis

Ten grams of tomato fruits were homogenized and then adjusted a volume by 0.4% oxalic acid to 100 ml. The samples were filtered and taken 10 ml to be titrated with 2,6 dichlorophenol-indophenol (*AOAC, 2000*).

## Antioxidant activity analysis

The 1,1-diphenyl-2-picrylhydrazyl (DPPH) free-radical scavenging assay, as described by *Brand-Williams, Cuvelier & Berset (1995)*, was conducted to evaluate the antioxidant capacity of tomato fruit extracts. For this assay, a stock solution was prepared by dissolving 16 mg of DPPH in 100 mL of ethanol. In a test tube, a mixture containing 1.8 mL of solutions (DPPH, Tris buffer, and 85% ethanol) was combined with 600 µL of the tomato fruits extract. A standard solution of 0.4 mM Trolox in 15% ethanol was used for comparison. Following the addition of the tomato fruit extract, the tubes were stored in complete darkness for 30 min to allow the reaction to proceed. The absorbance of the mixture was then measured at a wavelength of 525 nm. The percentage of DPPH scavenging effect was calculated using a specific formula and the results were compared against a standard curve to determine the antioxidant capacity of the extracts. This method provides a quantitative measure of the ability of the tomato fruit extracts to neutralize free radicals, indicating their antioxidant potential. The calculation of DPPH was shown below:

$$\% \text{ of DPPH scavenging effect} = [(A_0 - A_s) \div A_0] \times 100$$

where: $A_0$- absorbance of control; As- absorbance of sample.

## Lycopene analysis

The extraction and quantification of lycopene followed the method outlined by *Serino et al. (2009)*. This process began with an assay sample of approximately 600 mg being treated with a mixture of 5 mL acetone, 5 mL ethanol, and 10 mL n-hexane for 15 min to facilitate extraction. After this period, the mixture was decanted to separate the upper n-hexane phase, which was then evaporated under nitrogen to prevent oxidation and degradation by light. The dry residue left after evaporation was dissolved in 1,250 μL of MSolv, a solvent mixture comprising ethyl acetate (EA), dichloromethane (DCM), and hexane (Hex) in the ratio of 80:16:4 (v/v/v). This procedure ensured the final extract had a volume and solvent composition compatible with the micromethod intended for high performance liquid chromatography (HPLC) analysis.

After preparation, the extract was thoroughly homogenized using a vortex, filtered through a PTFE filter (0.45 μm), and either analyzed immediately with HPLC or stored at −20 °C until the time of assay. The HPLC assay targeted a wavelength range of 474 nm, specifically chosen for the detection and quantification of lycopene content in the tomato fruit samples.

## Statistical analysis

The data were analyzed using a one-way analysis of variance (ANOVA) to determine the effect of the four treatments, and multiple comparisons between treatment means were conducted using the least significant difference (LSD) test at a 95% confidence level.

# RESULTS

## Root colonization in tomato seedling

After the inoculation of actinomycetes into the root zone of tomato seedlings, seedlings from each treatment group were analyzed for root colonization using Scanning Electron Microscopy (SEM: JSM IT300). It was observed that TGsR-03-04, TGsL-02-05, and the combined treatment of TGsR-03-04 + TGsL-02-05 successfully colonized the roots of the tomato plants, as shown in Fig. 1.

## Effect of endophytic actinomycetes on height of tomato plant at 14, 28, 56 and 112 DAT

After transplanting, the growth of tomato plants was recorded at 14, 28, 56, and 112 days after transplanting (DAT), as depicted in Fig. 2. The results indicate that the inoculation with different endophytic actinomycetes significantly affects the height of tomato plants. At 14 DAT, the plants treated with TGsR-03-04 achieved the highest average height of 19.40 cm. In contrast, the lowest average height at this stage was observed in plants inoculated with a combination of TGsR-03-04 + TGsL-02-05, measuring 16.40 cm. However, by 28 DAT, there was no significant difference in the height of tomato plants across the treatments. Subsequent observations at 56 and 112 DAT revealed that the control treatment (without any inoculation) resulted in the tallest plants, with average heights of 74.93 cm and 96.73 cm, respectively. Conversely, the treatment that resulted in the shortest plants was the one

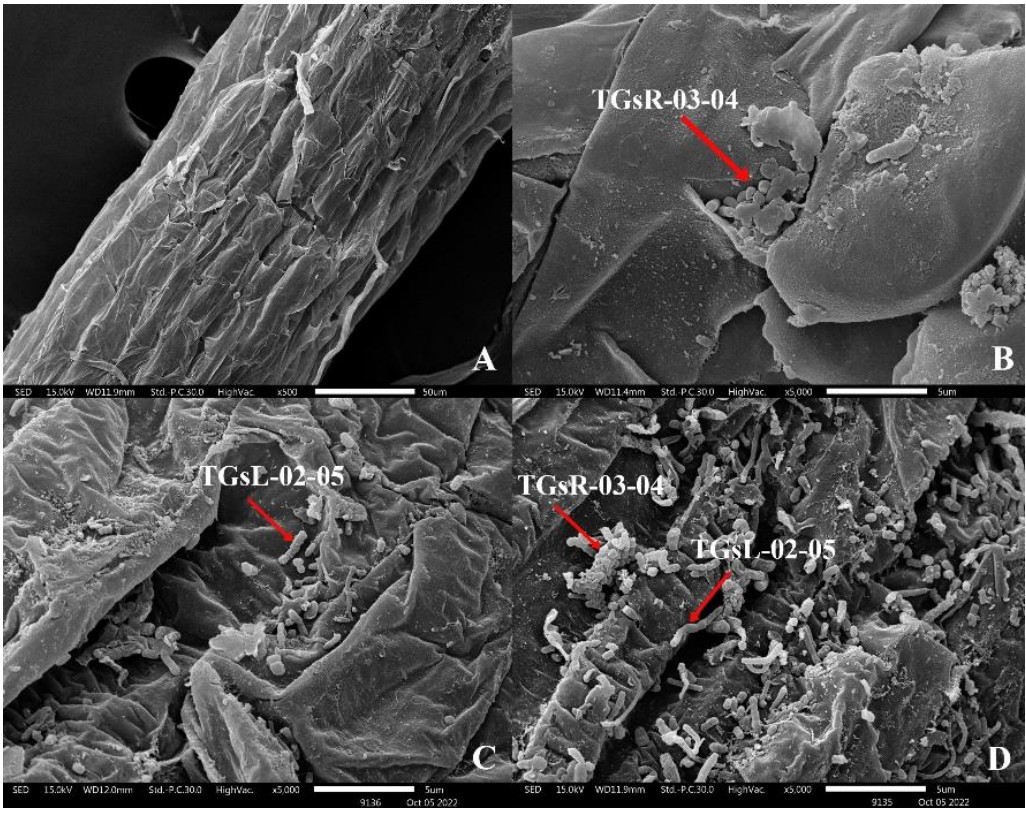

**Figure 1** Characteristics of root colonization of (A) control, (B) *Streptomyces violaceorectus* (TGsR-03-04), (C) *Nocardiopsis alba* (TGsL-02-05) and (D) *Streptomyces violaceorectus* (TGsR-03-04) with *Nocardiopsis alba* (TGsL-02-05).

inoculated with TGsL-02-05, which had average heights of 62.27 cm and 85.87 cm at 56 and 112 DAT, respectively.

## Effect of endophytic actinomycetes on yield and yield quality in tomato fruits

The results indicated that while the inoculation of different endophytic actinomycetes did not affect the number of fruits per plant, fruit width, or Brix value, it did have an impact on the total yield per plant, weight per fruit, and fruit length. The treatment inoculated with TGsL-02-05 resulted in the highest yield, reaching 1,471 g per plant. Conversely, the combined inoculation with TGsR-03-04 and TGsL-02-05 produced the lowest yield per plant, at 1,251 g. Furthermore, the treatment with TGsR-03-04 alone was notable for producing the longest fruits and the highest weight per fruit, measuring 52.37 mm in length and 47.38 g in weight, as detailed in  Table 2 and Fig. 3.

## Effect of endophytic actinomycetes on vitamin C, antioxidant and lycopene in tomato fruits

The results indicated that the inoculation of different endophytic actinomycetes significantly influenced the vitamin C and lycopene content in tomato fruits, as shown

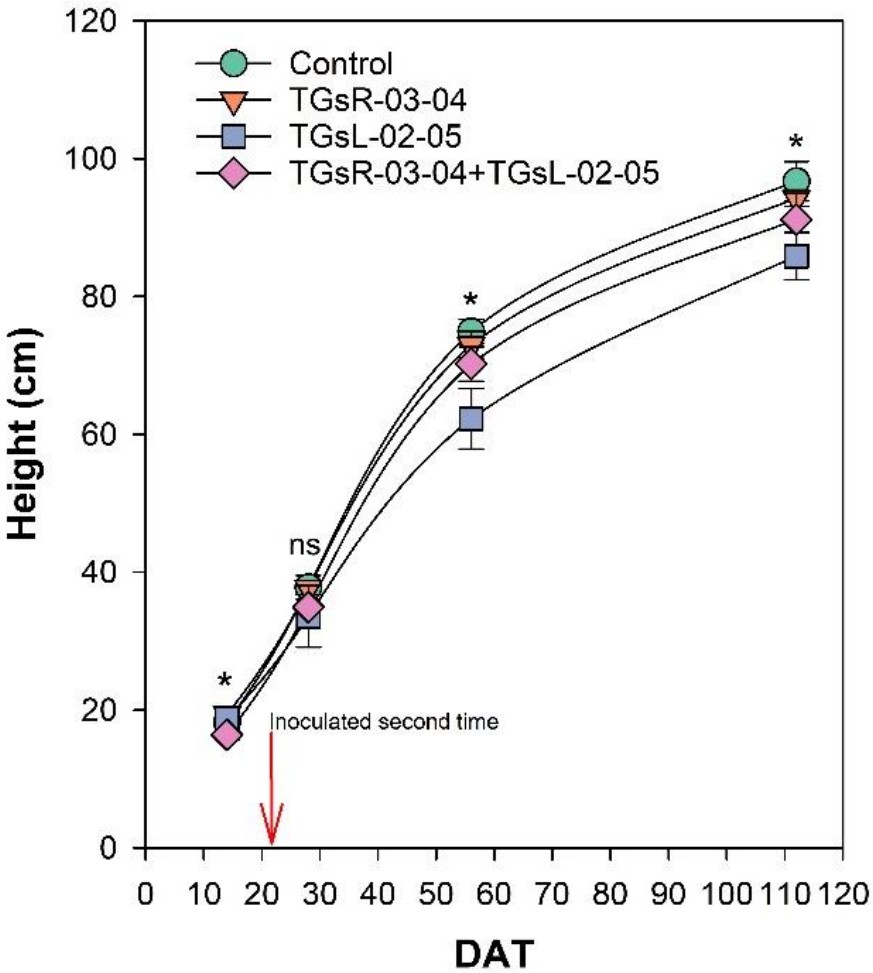

**Figure 2   Effect of endophytic actinomycetes on height of tomato plant at 14, 28, 56 and 112 days after planting (DAT).** The asterisk indicates significant statistical differences based on ANOVA ($p < 0.05$).

in Fig. 4. Specifically, the inoculation with TGsR-03-04 led to a significant increase in vitamin C, with levels reaching 23.30 mg per 100 g, which was higher than those observed in treatments with TGsL-02-05 (12.25 mg per 100 g), the combination of TGsR-03-04 and TGsL-02-05 (13.67 mg per 100 g), and the control (17.11 mg per 100 g). In terms of lycopene content, the inoculation with TGsR-03-04 resulted in the highest value of 145.92 μg/g. This was followed by a combination treatment of TGsR-03-04 with TGsL-02-05, which yielded 131.80 μg/g. The control treatment showed a lycopene content of 80.02 μg/g, while treatments with TGsL-02-05 alone resulted in the lowest value of 77.64 μg/g.

However, the inoculation with different endophytic actinomycetes did not have a discernible impact on the antioxidant activity in the tomato fruits. The antioxidant activity was found to range between 98.21–103.62 μmol Trolox Equivalent (TE) per g of sample, indicating that while there was a trend towards higher antioxidant activity in tomatoes

**Table 2** Yield (g), weigh/fruits (g), number/plant, fruit width (mm), fruit length (mm) and brix (%) of tomato fruits under different inoculation endophytic actinomycetes.

| Treatment | Yield (g plant$^{-1}$) | Yield Components | | | | |
|---|---|---|---|---|---|---|
| | | Weigh per fruits (g) | Number per plant | Fruit width (mm) | Fruit length (mm) | Brix (%) |
| Control | 1,455 ± 23[a] | 43.64 ± 0.6[ab] | 41.93 ± 5.3 | 39.50 ± 1.0 | 51.46 ± 1.0[ab] | 6.71 ± 0.1 |
| TGsR-03-04 | 1,323 ± 45[b] | 47.38 ± 1.9[a] | 42.67 ± 2.9 | 40.26 ± 0.8 | 52.37 ± 1.4[a] | 6.92 ± 0.4 |
| TGsL-02-05 | 1,471 ± 35[a] | 41.86 ± 0.4[b] | 52.10 ± 4.2 | 38.63 ± 0.5 | 48.74 ± 0.8[b] | 6.99 ± 0.1 |
| TGsR-03-04+ TGsL-02-05 | 1,251 ± 35[b] | 41.88 ± 1.2[b] | 46.00 ± 1.5 | 39.17 ± 1.2 | 49.16 ± 0.5[ab] | 6.68 ± 0.4 |
| *F-test* | * | * | ns | ns | * | ns |
| *CV* | 4.63 | 4.96 | 12.02 | 4.55 | 3.51 | 8.13 |

**Notes.**
*Shows significant difference at $P < 0.05$.

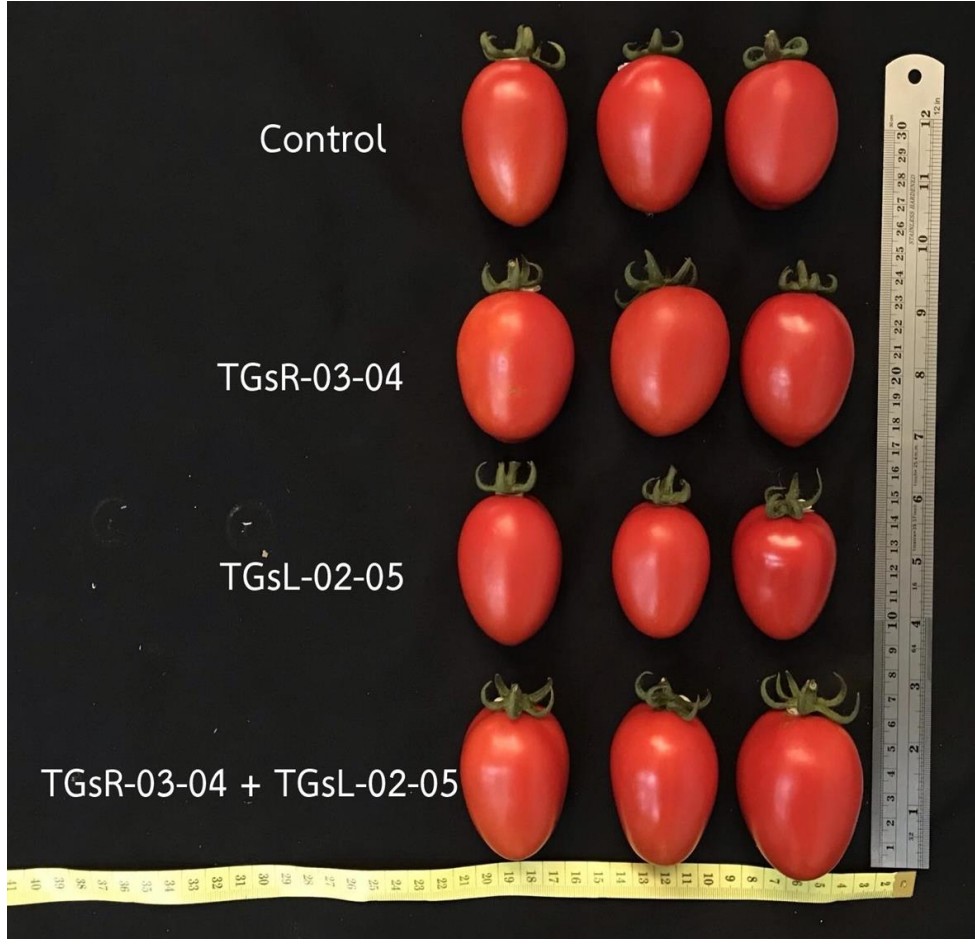

**Figure 3** Tomato fruits characteristics from each treatment.

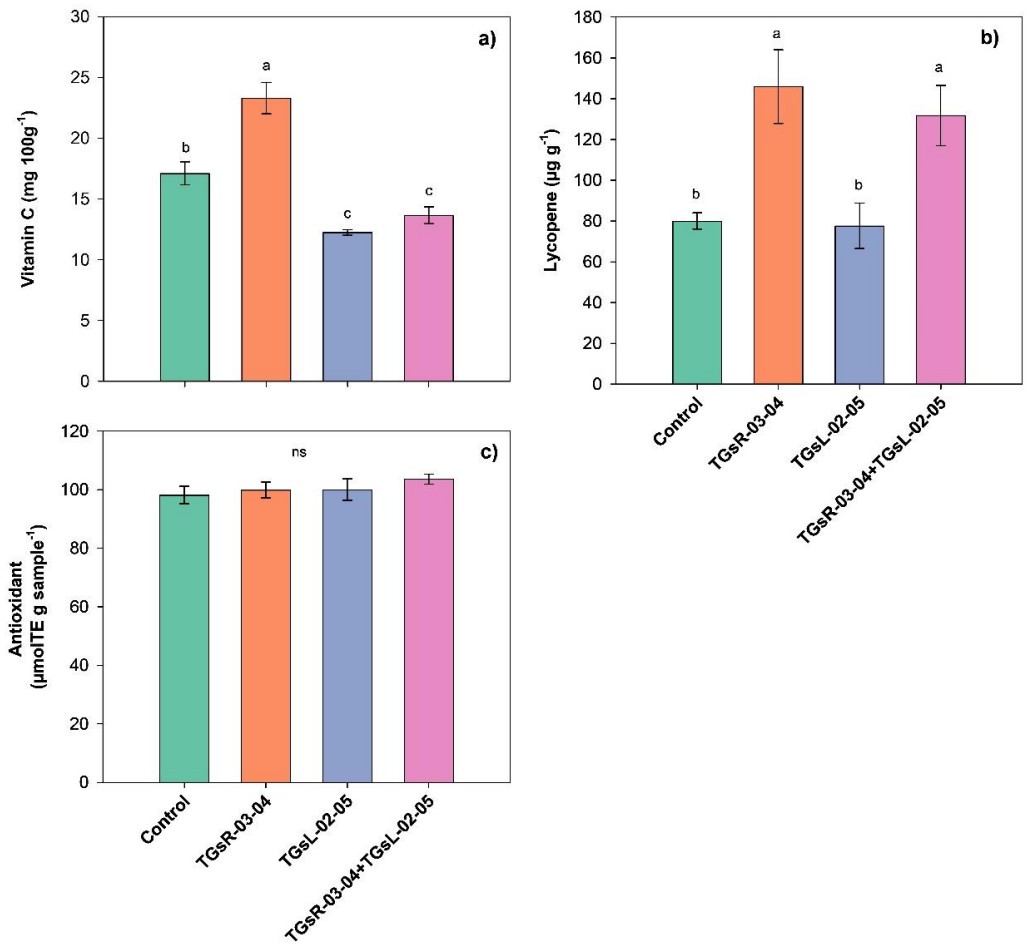

**Figure 4** **Effect of endophytic actinomycetes on vitamin C content (A), lycopene content (B) and antioxidant activity (C) in tomato fruits at second harvest.** Noted: Error bar indicates standard error. Different letters in column imply significant difference at $p \leq 0.05$ and ns; not - significant.

inoculated with both TGsR-03-04 and TGsL-02-05 compared to the control, the differences were not statistically significant ($P > 0.05$).

## Influence of endophytic actinomycetes on tomato growth and yield quality explained by principal component analysis

Figure 5 suggests that PC1 and PC2 together explained a significant portion of the total variance in your dataset, 62.08%. The high positive loading scores for plant height, fruit length, fruit width, and weight per fruit on PC1 suggest that these variables had a strong positive correlation with the variance explained by PC1. Conversely, the number of fruits per plant and total yield had lower negative loading scores, indicating a negative correlation with PC1.

The treatment with TGsR-03-04 seemed to have a marked influence on the weight, length, and width of the fruits, suggesting that this treatment may contribute to larger fruit size. In Fig. 6, the effect of TGsR-03-04 on vitamin C content was noted, while the

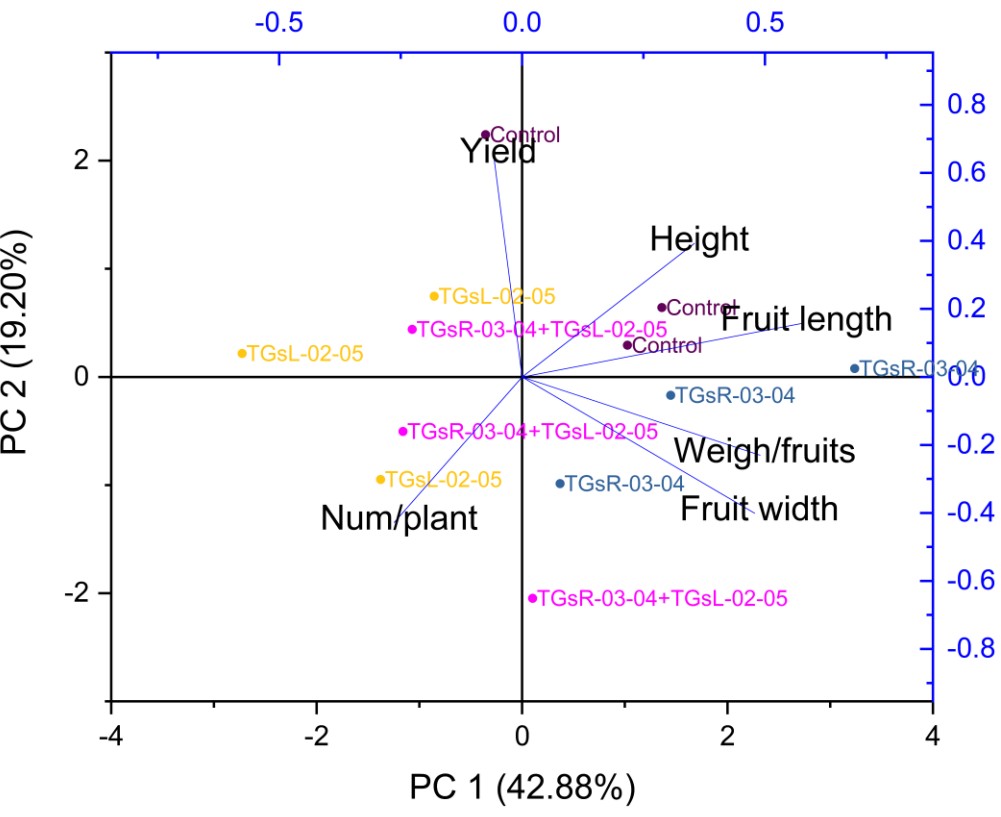

**Figure 5** Principal component analysis (PCA) graphics of interaction for yield and fruit quality (height, fruit length, fruit width, weigh/fruits, number/plant) under difference inoculation with endophytic actinomycetes.

combined treatment of TGsR-03-04 and TGsL-02-05 seemed to influence the antioxidant levels.

The negative correlations mentioned, where yield showed an inverse relationship with both antioxidant activity and lycopene content, were particularly intriguing. This could imply that conditions or treatments that lead to increased yield may do so at the expense of these particular phytonutrients.

The significant correlations between weight per fruit, fruit length, fruit width, and vitamin C (as shown in Fig. 7) support the idea that larger fruits, which were likely heavier and have greater dimensions, also tended to have higher vitamin C content.

## DISCUSSION

This study revealed a clear significant effect of endophytic actinomycetes on tomato yield quality, but it did not affect growth of tomato. For tomato growth promoting effect, the results did not seem so clear since the control had a better growth in term of height development. Tomato growth was measured at each recorded growth stage (14, 28, 56, and 112 days after transplantation, DAT). It was found that at 14 DAT, plants treated with

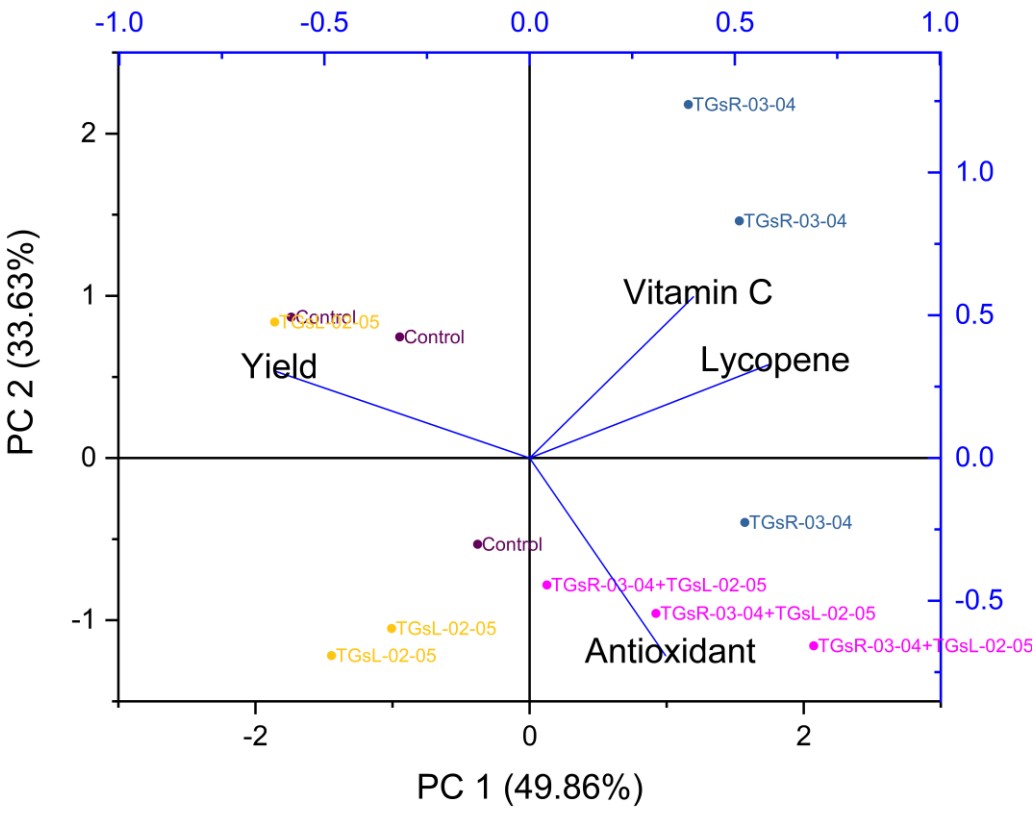

**Figure 6** Principal component analysis (PCA) biplots of interaction of yield and vitamin C, antioxidant, lycopene under difference inoculation with endophytic actinomycetes.

TGsR-03-04 achieved the greatest height with 19.40 cm. Similarly, *Shahzad et al. (2021)* studied on the potential of PGPR for promoting tomato growth. Tomato height at 14 DAT found that when applied *Bacillus aryabhattai* (SRB02) on tomato plants, the height of tomato was improved by 37.4% as compared to the control plants. However, at 28, 56, and 112 DAT, the control treatment had the highest growth in terms of height, which may be the result of the tomato plants being in the flowering and fruiting stages at this time. This may have caused the actinomycetes to promote growth in other parts of the plant instead, such as fruit components and substances. From this experiment, the accumulation of nitrogen in tomato fruit in the inoculated treatments was significantly higher than in the control treatment. For example, the TGsR-03-04 inoculated treatment (2.28%) gave higher nitrogen values than the control treatment (2.18%). On the other hand, *Loganathan et al. (2014)* had shown a potential of PGPR (*Bacillus amyloliquefaciens* isolate BA1 and *Bacillus subtilis* isolate BS2) for producing phytochemicals like indole-3-acetic acid (IAA) and siderophores that can promote tomato growth at 45 DAT. The highest height was found in BS2 (57.7 cm) followed by BA1 (51.95 cm), the least height of tomato was in control (no inoculation) (44.80 cm). This significant positive impact of PGPR on tomato growth aligns with findings from other research. *Gashash et al. (2022)* discovered that

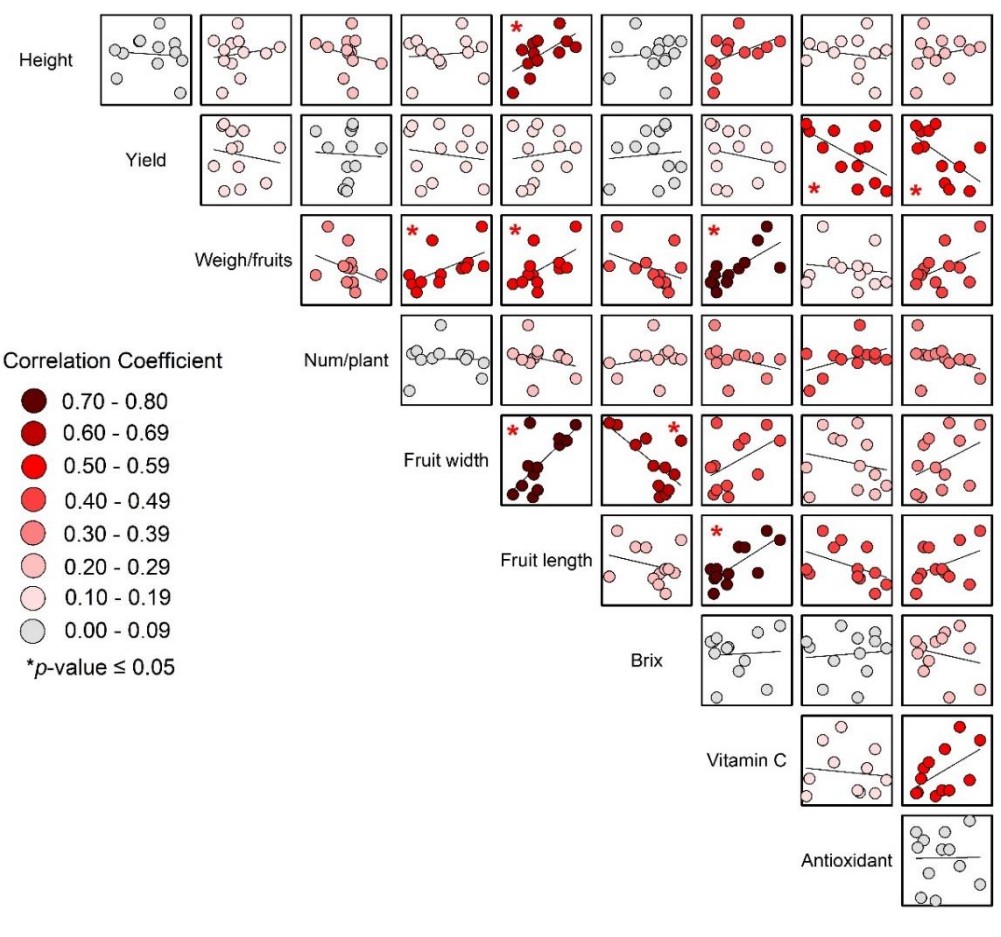

**Figure 7 Correlations of yield and yield quality of tomato.**

inoculating tomato with microorganisms capable of acting as PGPR led to notable results. Specifically, the treatment that involved inoculation with a combination of *Bacillus subtilis* and *Bacillus amyloliquefaciens* resulted in the tallest tomato plants, reaching a height of 117 cm. Microorganisms capable of producing substances that promote plant growth, known as PGPR, can synthesize Indole-3-acetic acid (IAA), which is implicated in plant growth processes (*Ahemad & Kibret, 2014*; *Afzal, Shinwari & Iqrar, 2015*). This production of IAA is directly linked to plant growth (*Ahemad & Kibret, 2014*) and facilitates improvement in plant growth by stimulating the hormone auxin (*Ruzzi & Aroca, 2015*). Similarly, *Klangsawad & Thummabenjapone (2016)* reported that plants treated with inoculations showed greater heights at 14, 28, and 40 DAT compared to those without inoculations, attributed to the production of plant growth regulators in the auxin group. The auxin produced by microorganisms affects plant growth (*Khare & Arora, 2010*). It is suggested that in our study, the endophytic actinomycetes did not promote growth in terms of tomato

height, as there was no interaction with any actinomycete isolate. This correlation is shown in Fig. 4.

In our study, the results demonstrated that inoculation treatment with TGsL-02-05 (*Nocardiopsis alba*) tended to produce the highest number of fruits per plant (52.10 fruits). This finding aligns with the work of *Elsharkawy et al. (2023)*, which discovered that *Pseudomonas putida* inoculation yielded the most fruits per plant (43.5 fruits), while the control resulted in the lowest fruits (20.6 fruits). Furthermore, TGsL-02-05 (*Nocardiopsis alba*) inoculation also led to the highest yield per plant (1.47 kg), while the lowest yield was recorded for the treatment inoculated with TGsR-03-04 (*Streptomyces violaceorectus*) along with TGsL-02-05 (*Nocardiopsis alba*), producing 1.25 kg per plant.

On a different note, *Gowda et al. (2023)* investigated rhizobacteria in greenhouse conditions, finding that co-inoculation with *Staphylococcus sciuri*, *Bacillus pumilus*, and *Priestia megaterium* resulted in the highest yield (624 g/plant) compared to single inoculation and control treatments. This suggests that a synergistic effect from multiple microorganisms can be more effective in promoting growth, as noted by *Wang et al. (2023)*. Our results also highlighted that TGsR-03-04 (*Streptomyces violaceorectus*) inoculation resulted in significantly longer (52.37 mm) and heavier fruits (47.38 g), with a tendency to increase fruit width (40.26 mm). This supports findings by *Sreeja & Gopal (2013)*, who observed that actinomycete endophytes could lead to heavier tomato fruits compared to controls, underscoring the growth-promoting capability of endophytic actinomycetes (*Pillay & Nowak, 1997*). Nitrogen, a crucial factor for fruit size, was found to be more concentrated in the fruits treated with TGsR-03-04 (2.28%) than in the control (2.18%), indicating its role in reproductive system development due to its presence in proteins and amino acids (*Sainju, Dris & Singh, 2003*). Furthermore, soluble solids content (brix) was higher in tomatoes inoculated with TGsL-02-05 (*Nocardiopsis alba*) (6.99%) compared to the control (6.71%). *Katsenios et al. (2021)* also reported higher brix values in tomatoes inoculated with PGPR bacteria, specifically *B. pumilus* (4.75%) *versus* the control (4%), highlighting PGPR's role in enhancing yield quality. This is particularly relevant in the tomato industry, where higher brix values are indicative of superior product quality (*Maach et al., 2020*).

The impact of endophytic actinomycetes on the nutritional content of tomato fruits, including vitamin C, antioxidants, and lycopene, highlights the potential for these microorganisms to enhance fruit quality significantly. The treatment involving inoculation with TGsR-03-04 (*Streptomyces violaceorectus*) resulted in tomato fruits with the highest vitamin C content, underlining the capacity of endophytic actinomycetes, particularly those from the Streptomyces genus, to produce secondary metabolites with a broad range of biological activities (*Solecka et al., 2012*; *Palaniyandi et al., 2013*; *Qin et al., 2011*). This finding is in agreement with *Ordookhani et al. (2012)*, who noted that PGPR bacteria *Pseudomonas* inoculation led to increased vitamin C content in tomato fruits, attributed to the PGPR's role in promoting plant growth through hormone production and the solubilization of beneficial nutrients *via* secondary metabolites (*Gashash et al., 2022*).

Moreover, the lycopene content was highest in tomatoes treated with TGsR-03-04, aligning with research by *Singh & Pandey (2021)*, which showed that inoculation with

endophytic bacteria increased lycopene content in hybrid tomatoes. This enhancement in lycopene content is thought to be related to the improved plant health and growth facilitated by PGPR, including nutrient solubilization and increased availability of plant nutrients like nitrogen, which plays a crucial role in many plants metabolic processes (*Abushita, Daood & Biacs, 2000*; *George et al., 2004*; *Albornoz, 2016*).

Additionally, the experiment revealed that tomatoes inoculated with a combination of TGsR-03-04 and TGsL-02-05 exhibited a trend towards higher antioxidant activity compared to the control (non-inoculated with actinomycetes). This increase in antioxidant activity is likely connected to the higher lycopene content observed with PGPR inoculation, as suggested by *Andreou et al. (2020)*, who found that PGPR-inoculated antioxidants were related to carotenoid compounds, particularly lycopene. This relationship underscores the synergistic effects of utilizing multiple microorganisms for inoculation, which not only enhances plant growth but also improves the nutritional quality of the fruits, potentially offering a sustainable approach to boosting the health benefits of tomatoes.

## CONCLUSIONS

The findings of this research demonstrate that different types of actinomycete endophytes possess the capability to enhance the yield, quality, and content of beneficial substances in tomatoes, with variations observed across different plant growth stages. Specifically, the actinomycete endophyte TGsR-03-04 (*Streptomyces violaceorectus*) has been shown to increase fruit size and the levels of vitamin C and lycopene in tomato fruits. On the other hand, TGsL-02-05 (*Nocardiopsis alba*) contributes to the overall yield of tomato plants and enhances the nutritional and bioactive compound content in the fruits. Consequently, these actinomycete endophytes hold promising potential for application as bio-fertilizers in tomato cultivation systems, offering a sustainable approach to improve both the quantity and quality of tomato production in the future.

### Funding

This research was funded by Research Grant for Agriculture and Agro-Industry from Agricultural Research Development Agency (Public Organization) of 2022. Moreover, this research work was also supported by Graduate School, Teaching Assistant and Research Assistant Scholarships (TA/RA) of 2020-2022, Chiang Mai University. The funders had no role in study design, data collection and analysis, decision to publish, or preparation of the manuscript.

### Grant Disclosures

The following grant information was disclosed by the authors:
Research Grant for Agriculture and Agro-Industry from Agricultural Research Development Agency (Public Organization) of 2022.
Graduate School, Teaching Assistant and Research Assistant Scholarships (TA/RA) of 2020-2022, Chiang Mai University.

## Competing Interests

The authors declare there are no competing interests.

## Author Contributions

- Jeeranan Khomampai conceived and designed the experiments, performed the experiments, analyzed the data, prepared figures and/or tables, authored or reviewed drafts of the article, funding acquisition, and approved the final draft.
- Nakarin Jeeatid conceived and designed the experiments, prepared figures and/or tables, authored or reviewed drafts of the article, and approved the final draft.
- Thewin Kaeomuangmoon conceived and designed the experiments, prepared figures and/or tables, and approved the final draft.
- Wasu Pathom-aree performed the experiments, prepared figures and/or tables, and approved the final draft.
- Pharada Rangseekaew performed the experiments, authored or reviewed drafts of the article, and approved the final draft.
- Thanchanok Yosen performed the experiments, authored or reviewed drafts of the article, and approved the final draft.
- Nuttapon Khongdee analyzed the data, prepared figures and/or tables, authored or reviewed drafts of the article, and approved the final draft.
- Yupa Chromkaew conceived and designed the experiments, performed the experiments, analyzed the data, prepared figures and/or tables, authored or reviewed drafts of the article, and approved the final draft.

## Data Availability

The raw data is available in the Supplementary File.

## Supplemental Information

Supplemental information for this article can be found online at http://dx.doi.org/10.7717/peerj.17725#supplemental-information.

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
