# Peer review of "Endophytic actinomycetes promote growth and fruits quality of tomato (Solanum lycopersicum): an approach for sustainable tomato production"

_PeerJ, doi:10.7717/peerj.17725_

## Round 0.1 · original submission · Major Revisions

A major revision has been recommended by experts. Please revise the article considering reviewers suggestions.

·

Basic reporting

Author should submit data pertaining to plant growth promotion traits of actinomycetes at least as supplementary material.

Experimental design

No commenrts

Validity of the findings

No comments

Additional comments

No comments

Reviewer 2 ·

Basic reporting

The article is dealt with the effect of endophytic actionmycetes in enhancing the tomato growth and fruit quality. In the current scenario, the development of microbial bio-agents is gaining importance specifically for the sustainable management of soil and plant health. Though, several studies has been reported for Bacillus and derived genera, the exploration of actinomycetes is still in the nascent stage. In this context, the reported study adds scientific knowledge on the subject.

The following are the point-wise comments to the authors to address.
1.In the introduction section, the authors have mentioned that the excessive use of chemicals lead to decline in the soil and plant performance and the plant growth promoting microorganisms is helpful to reduce the use of chemicals in tomato cultivation(line no. 58-60). However, the same is missing in the hypothesis statement.
2.There are grammatical mistakes in the manuscript, for example, line no. 119. and Line no. 277. Reorganize the sentences. Check the grammar and sentences throught the manuscript.
3.The uniform unit of measurement should be given. For example line no. 141 and 142, weight is given in different forms (grams/g).

Experimental design

1.The experimental design fails to address the partial replacement of chemicals with plant growth promoting actinomycetes.The nutrients supplementation to the crop plants are given evenly for both inoculated and un-inoculated plants. The authors itself mentioned that the actinomycetes isolates used in the study have growth and nutrient supplementation properties (line no. 112-113). And, the major purpose of the study is to find an alternative to chemicals in tomato cultivation and the same is not emphasized in this study.
2.Line no. 106- detail the control
3.:Line no. 111- whether the isolates were procured isolated in this study?. If isolated, the details of isolation and screening should be given.

Validity of the findings

1.Line no. 253: Check the value of fruit length (not verified in Table 2)
2. Line no. 294: As the endophytic actinomycetes does not have significant effect on growth of tomato plants, change the sentence.
3. Line no. 380: parts should be rather “stages”
4. The conclusion has controversial statement with reference to the findings. Therefore, the conclusion should be rewritten.

Additional comments

nil

Reviewer 3 ·

Basic reporting

The article has professional article structure and is clear and unambiguous for the readers. The English language is sufficient along with the literature references.

Experimental design

The research question of the paper is well defined and fills an identified gap. Methods were described with detail and are able to replicate.

Validity of the findings

No comment

Additional comments

The study underscores the effectiveness of endophytic actinomycetes in bolstering tomato growth and quality, thus tackling agricultural hurdles and advocating for sustainable farming methodologies. The revised version is significantly improved; however, there are minor edits needed.

1) Align the objectives with the conclusion through the Materials & Methods, Results, and Discussion sections for better coherence.
2) The references in the introduction seem somewhat dated. Consider incorporating newer references to enhance the relevancy and currency of the study.

·

Basic reporting

This study on the influence of endophytic actinomycetes on tomato growth and fruit quality holds significant importance in agricultural practices. By exploring alternative methods to chemical-based approaches, it offers potential solutions to declining yields and soil health issues. Moreover, its findings could pave the way for sustainable and environmentally friendly strategies in tomato cultivation, addressing both production challenges and consumer health concerns. Ultimately, this research contributes to the advancement of agricultural science and promotes healthier, more resilient food production systems. However, to ensure the manuscript meets the highest standards of scientific rigor and clarity, I recommend that the authors revise their submission to address the following major concerns:

How do endophytic actinomycetes contribute to the growth and health of tomato plants, and what specific mechanisms do they employ to enhance plant resilience and fruit quality?
In what ways does the excessive use of chemicals in tomato production impact plant nutrient availability and soil health, and how does this challenge necessitate alternative approaches, such as the use of microorganisms like actinomycetes?
What are the key objectives of the current study regarding the influence of endophytic actinomycetes on tomato growth and fruit quality, and how do these objectives address the gaps in existing research, particularly concerning the effects of Streptomyces and Nocardiopsis on tomato plants?
How did the colonization patterns of endophytic actinomycetes differ across the various treatment groups, as observed through Scanning Electron Microscopy analysis of tomato root systems?
What were the trends in tomato plant height over the course of the experiment (at 14, 28, 56, and 112 days after transplanting), and how did the different inoculation treatments influence these growth dynamics?
Can you elaborate on the specific effects of endophytic actinomycetes on yield and yield quality parameters of tomato fruits, including total yield per plant, weight per fruit, fruit length, vitamin C content, and lycopene content, as revealed by the experimental results?
How did the colonization patterns of endophytic actinomycetes differ across the various treatment groups, as observed through Scanning Electron Microscopy analysis of tomato root systems?
What were the trends in tomato plant height over the course of the experiment (at 14, 28, 56, and 112 days after transplanting), and how did the different inoculation treatments influence these growth dynamics?
Can you elaborate on the specific effects of endophytic actinomycetes on yield and yield quality parameters of tomato fruits, including total yield per plant, weight per fruit, fruit length, vitamin C content, and lycopene content, as revealed by the experimental results?

The English quality of the text is generally good, with clear communication of scientific concepts and findings. However, there are some instances of grammatical errors, awkward phrasing, and typographical issues that could be improved for better clarity and readability. For example:

"However, at subsequent measurements (28, 56, and 112 DAT), the control treatment plants exhibited the highest growth in terms of height." (This sentence could be rephrased for smoother readability)
"Tomatoes inoculated with a combination of TGsR-03-04 and TGsL-02-05 exhibited a trend towards higher antioxidant activity compared to the control." (Consider clarifying what "the control" refers to)
"This finding aligns with the work of Elsharkawy et al. (2023), which discovered that Pseudomonas putida inoculation yielded the most fruits per plant (43.5 fruits), contrasting with the lowest fruit count found in the control group (20.6 fruits)." (The phrase "contrasting with the lowest fruit count found in the control group" could be better integrated into the sentence for smoother flow)
Overall, while the text effectively conveys scientific information, refining the language for clarity and coherence would enhance its quality further.

Experimental design

No comments

Validity of the findings

No comments

Additional comments

None

---

## Round 0.2 · accepted · Accept

Reviews have accepted your article as they are satisfied with your response and revised article.

Reviewer 3 ·

Basic reporting

I have no further suggestions and believe the authors have significantly improved the manuscript, making it now suitable for publication.

Experimental design

I have no further suggestions and believe the authors have significantly improved the manuscript, making it now suitable for publication.

Validity of the findings

I have no further suggestions and believe the authors have significantly improved the manuscript, making it now suitable for publication.

·

Basic reporting

I have no further suggestions. The revised manuscript is now suitable for publication.

Experimental design

No comment

Validity of the findings

No comment

Additional comments

No comment